# Influence of PVB Interlayer Mechanical Properties on Laminated Glass Elements Design in Dependence of Real Time-Temperature Changes

**DOI:** 10.3390/polym14204402

**Published:** 2022-10-18

**Authors:** Josip Galić, Lucija Stepinac, Antonia Bošnjak, Ivana Zovko

**Affiliations:** Faculty of Architecture, University of Zagreb, 10000 Zagreb, Croatia

**Keywords:** laminated glass, foil, PVB, static test, viscoelasticity, temperature-time superposition, Wölfel-Bennison approach, relaxation modulus

## Abstract

Most used laminated glass is composed of float glass plies bonded together with a viscoelastic Polyvinyl Butyral (PVB) interlayer. The shear stiffness of the polymeric interlayer is the key factor in the behavior of laminated glass. Structural engineers in the past were designing laminated glass regardless of the shear coupling of the plies. This approach with a high level of reliability led to expensive laminated glass structures due to insufficient knowledge of foil properties. Most of the current standards suggest methods that consider the shear coupling of the plies. This paper presents the experimental data from a static loading test performed on a laminated glass panel exposed to changing temperatures. The deformations were observed for 48 h. The measured results were compared with the known analytical design approaches and in addition with the finite element modeling (FEM) analysis in the available software for laminated glass design. A simplified design approach that simulates foil behavior in dependence on load duration and temperature change was adopted in this study. Design approaches that use effective thickness calculations are used with the Young and shear relaxation modulus provided by the foil producer. The imprecision of the Eurocode standards for glass design, and the propensity to change the approach to the calculation by introducing more precise parameters were expounded. The results when combining the time-temperature superposition (TTS) and the Wölfel–Bennison approach were found to be in very good agreement with the FEM analysis of 3D solid elements in Abaqus and measured data.

## 1. Introduction

Laminated glass is a high-quality product whose application is constantly increasing. Simple design calculations exist in guidelines and manuals. However, more precise calculations that take into account more factors are needed for the design of laminated glass. Vibration and noise isolation, impact resistance, and mitigation of post-fracture glass fallout are the advantages of laminated glass [1]. Laminated glass is a composite of two or more glass layers with a thin elastomeric interlayer, which affects the degree of their coupling. The shear stiffness of the polymeric layer is the key factor in the behavior of laminated glass.

Structural engineers in the past were designing laminated glass regardless of the shear coupling of the plies. This approach with a high level of reliability leads to expensive laminated glass structures due to insufficient knowledge about the foil’s properties. Most of the current standards imply methods that consider the shear coupling of the plies. Properties of PVB interlayers, more precisely the relaxation shear modulus, are given in data sheets from static loading tests regarding temperature and loading rate. This enables a more precise design of laminated glass structures for structural engineers. 

The load-bearing capacity of laminated glass is difficult to calculate, and this is the reason for its limited effective use. The stiffness and load-bearing capacity of laminated glass depends on, among others, the shear connection of its glass plies, which is achieved using foil made of different polymer materials. Because of this complex behavior of laminated glass, simplified design rules are often applied in practice. The simplification of the design lies in the assumption that the behavior of the polymer is linearly elastic with a secant shear modulus that considers the load duration and ambient temperature. In addition, geometrical nonlinearities are important from the aspect of the slenderness of the glass panel, but they are negligible when the panel is subjected to a transverse load only and in-plane forces are absent [2].

Material properties of laminated glass components have a wide spectrum. Glass has a Young’s modulus of approximately 70 GPa, while a value for the interlayer is dependent on the type of polymer, temperature, load type, i.e., static or dynamic [3], and the load duration. The secant shear modulus of the interlayer varies from 0.01 MPa to 300 MPa [2]. The creeping/relaxation property of the polymer interlayers under long-term loading will result in the shear modulus value decreasing [4]. Viscoelastic properties of the interlayer may result in a drop of effective shear modulus almost to the 0 MPa when the laminated glass is exposed to higher temperatures or when there is a constant loading under a longer period. References for several interlayer product types can be presented in the producer’s technical data forms, such as [5]. An Italian study by Biolzi et al. [6] investigated the response of laminated glass with PVB interlayer under long load duration at different temperatures and the effect of relative humidity was observed. The experiment showed how the effective shear modulus of PVB foil decreases significantly for temperatures higher than 30 °C and at 50 °C it had an extremely low value. For temperatures below 0 °C, the shear modulus was greater than 0.5 MPa and showed a slow decline in value even for long load duration. As for variation of the relative humidity, the experiment showed it has no significant influence on the shear modulus of the interlayer [6].

The bending stiffness of laminated glass is found to be lower than the stiffness of monolithic glass with the same thickness in a very short time after being mounted [7]. Monolithic and laminated glass behave similarly under lateral loads of shorter duration (for example, wind loads) at room temperature or below. Observable different behavior from the monolithic glass under short-term lateral pressures is not clearly defined, but it occurs around 49 °C [8]. Similar behavior of both glasses with the same nominal thickness under long-term lateral pressures (e.g., snow loads) occurs at temperatures of 0 °C or below when the foil is stiffer [8]. At elevated temperatures (approximately 77 °C), the behavior changes when the foil transits from a glassy to a rubbery state and the glass units start to behave like two separate layers without a bond [7].

Change in the maximum load capacity is expected in a composite of glass and polymer because of their dissimilar mechanical properties under the loading [9]. Therefore, the presence of the plastic or viscoelastic interlayer causes a decrease in the bending stiffness of the laminated glass. This is in contrast with the requirement to withstand bending moments induced by wind pressure, snow loading, and self-weight acting over comparatively large spans [7].

Various kinds of polymers and other plastics are used as materials for the interlayers. Widely used interlayers are poly-vinyl-butyral (PVB), ionoplast (SentryGlas^®^), ethylene vinyl acetate (EVA), polyethylene (PE) [10], and thermoplastic polyurethane (TPU) [11].

This paper presents the experimental data regarding the laminated glass panel gained from static loading. The experimental campaign was performed at KFK Ltd. in Zagreb, Croatia. Deformation was measured for a better understanding of the foil shear modulus and its dependence on temperature under uniform stress. Self-weight loading was chosen for constant stress application during the measured time. The deflection was then calculated using analytical formulations using several approaches: normative document EN 16612 [12], enhanced effective thickness approach [13], Wölfel–Bennison approach [13], and approach developed for this study, Wölfel–Bennison approach with TTS (time-temperature superposition) principle implementation [14]. Furthermore, numerical 3D FEM analysis in software was obtained for the comparison of design approaches of laminated glass. A simulation was performed using software with laminated glass design options: Dlubal RFEM—RF GLASS [15] and SCIA Engineer [16] who implemented glass Addon followed by a Joint Research Centre Report [17] stipulating that prEN 16612 [12] and prEN 13474 [18] will be the basis of Eurocode for Glass structures. The European standard for glass is also compatible with DIN 18008 (National German Glass Design) [19]. According to the German glass standard DIN 18008 [19], the shear effect should be taken into account depending on the results. If the shear effect is favorable to the results, it must not be considered for the calculation and if the effect is unfavorable to the results, shear transfer, in its full capacity, must be considered [20].

Detailed solid FEM analysis through TTS (temperature-time shift) was performed in Abaqus SIMULIA [21]. Laminated glass deflection results are presented in this paper. Analysis was made by comparison of analytical, numerical, and experimental results. The purpose of this research is to understand the behavior of the laminated glass when the temperature and the load duration are variable, and how to consider all these parameters for a more accurate laminated glass design analysis. 

## 2. Materials and Methods

In trying to understand the bending stiffness of laminated glass, two extremes can be observed (Figure 1). With layered glass plies of equal thickness without the existence of the foil between them, plies can slide freely and the shear modulus tends to zero (Figure 1b). In Figure 1a, laminated glass is considered monolithic glass with a thickness equal to the sum of the thicknesses of the glass plies; glass plies are considered as absolutely bonded together and shear modulus tends to infinity. These two cases define the upper and lower stiffness limit of laminated glass: the case of layered glass plies determines the lower limit, and the case of monolithic glass the upper stiffness limit. The actual stiffness of laminated glass is within these borderline cases (Figure 1c) [13].

### 2.1. Theoretical Background

#### 2.1.1. Viscoelastic Response of the Interlayer

PVB foil is one of the first and still most commonly used interlayers for laminated glass and it is known as a viscoelastic material. Thus, structural engineers use a below-unity type factor in structural design considering that the PVB interlayer is not completely effective in transmitting shear forces between the layers of glass [7].

To determine the mechanical properties of viscoelastic materials [22], it is necessary to combine elastic behavior governed by Hooke’s law and viscous behavior governed by Newton’s law [23]. During the static test, two approaches are carried out: creep test and stress relaxation test (Figure 2).

For the creep test (Figure 2a), constant stress is applied to get strain propagation in time. During the stress relaxation test (Figure 2b) constant strain is applied to determine stress response in time. Under constant stress, it can be seen that after the instantaneous strain, the strain is gradually increasing over time, which is known as the creep effect. After removing the stress there is again the instant recovery but with the propagated viscous recovery (Figure 2a). When the constant strain is applied in the static test, there is an instantaneous stress response but with the relaxation effect, whereby the stress gradually decreases during the time under the constant strain (Figure 2b). During the dynamic test (Figure 3) stress and strain will vary over time in a sinusoidal manner but with a temporal offset δ between. The offset will vary depending on the viscosity of the material. For the materials closer to elastic mechanical behavior, the offset will be smaller. A temporal offset can be expressed through the phase angle *tan(*δ*) = E″/E′*, where *E′* is storage modulus and *E″* is loss modulus. Complex modulus *E** can be calculated using Equation (1), where the *“i”* is for imaginary unit $i=\sqrt{-1}$.
(1)$$E^{*}\left(\omega \right)=E^{\prime }\left(\omega \right)+iE^{\prime \prime }\left(\omega \right)$$

The combination of elastic solid and fluid behavior represented by springs and dashpots, respectively, allows to build linear viscoelastic models. The simplest model, based on relaxation curves is the Maxwell model which is widely used because of its relatively simple approach through the definition of the Prony series [24]. Prony series are defined using the process of fitting to experimental data. There are several methods for it [24], while the whole process can be applied using a homogenous logarithmic scale distribution inside of commercial program packages, such as Abaqus, Ansys, etc. Complex dynamics shear modulus can be expressed as Equation (2).
(2)$$G^{*}\left(\omega \right)=G^{\prime }\left(\omega \right)+iG^{\prime \prime }\left(\omega \right)$$

Expressions for the static modulus are presented in Equation (3), and for the dynamic modulus, Equations (4) and (5), where *a_T_* is the shift factor.
(3)$$G\left(t\right)=G\left(t/a_{T},T_{0}\right)$$
(4)$$G^{\prime }\left(\omega ,T\right)=G^{\prime }(a_{T}\overset{\wedge }{A}\cdot \omega ,T_{0}).\;$$
(5)$$G^{\prime \prime }\left(\omega ,T\right)=G^{\prime \prime }(a_{T}\overset{\wedge }{A}\cdot \omega ,T_{0})$$

Higher temperatures cause the mobility in polymer chains and therefore its stiffness decreases which cause a drop in the total stiffness of laminated glass composite. Within temperature changes, there is also time propagation that will affect the chain changes. The transition from glassy to rubbery phase can be described within temperature *T_g_ = T*_0_ (glass transition temperature). When the polymer is close to reaching its glass temperature, it is most affected by time and frequency. For temperatures below *T_g,_* the mobility in polymer chains is low and molecules cannot move freely. On the other hand, when the temperature is above *Tg*, molecules have significantly more freedom of movement, and as the temperature rises, their free volume does as well. As a result of the change in molecular mobility, that occurs throughout the glass transition interval, mechanical and physical properties of the interlayer change. There are several methods for determining the glass transition temperature (standardized specific volume measurement, DTA, DSC, DMA at a fixed cooling or heating rate), but the one described as superior by Li et al. [25] is the new TTS method. This method complies with time-temperature superposition (TTS) with a dynamic mechanical analyzer (DMA). 

Understanding the behavior of viscoelastic materials can be determined within the time and temperature-dependent superposition (TTS) principle [14]. The correlation between time and temperature is known as the TTS principle where temperature change will cause *E(t)* horizontal shift (Figure 4) on a logarithmic time scale [26]. TTS principle is based on the shift factors *a_T_* used for the time correction at a given temperature (*T*_1_, *T*_3_, *T*_4_). Equivalent relaxation time is expressed by *t/a_T_* and matched with the reference temperature *T_ref_* (*T*_2_).

Shift factor can be determined using Equation (6) proposed by the Williams, Landel, and Ferry (WLF) model [26]: (6)$$log(a_{T})=\frac{-C_{1}\left(T-T_{0}\right)}{C_{2}+\left(T-T_{0}\right)}$$
where *C*_1_ and *C*_2_ are the WLF constants that are dependent on the temperature in the exact moment *T* and reference temperature *T*_0_. The exact determination of *T_g_* is not possible, but according to ASTM D3418-97 [27], it can be concluded from the loss modulus *E″(**ω)*. For PVB foil, it is approximately +8 °C. Another value for reference temperature is recommended to use in the WLF approach [14]. Softening temperature *Ts =* 50 °C was taken as the reference temperature and WLF factors *C*_1_*^s^* = 8.86 and *C*_2_*^s^* = 101.6 in this research. Values for *C*_1_ and *C*_2_ may be used as general values proposed by William, Landel, and Ferry, or can be evaluated by Expressions (7) and (8):(7)$$C_{1}^{i}=\frac{C_{1}^{0}\;C_{2}^{0}}{\left(C_{2}^{0}+T_{i}-T_{0}\right)}$$
(8)$$C_{2}^{i}=C_{2}^{0}+T_{i}-T_{0}$$

Even though the TTS principle may be applied for all tested temperatures, the WLF method is applicable only for temperatures above *Tg* [28]. For temperatures lower than *Tg*, the shift factor $\alpha_{T}$ is recommended to be calculated using the Arrhenius activation energy in Equation (9), which is applicable only when the *T < Tg* [29].
(9)$${log}_{10}\alpha_{T}=\frac{E_{a}}{2.303R}\left(\frac{1}{T}-\frac{1}{T_{g}}\right)$$
where *E_a_* (J/mol) represents the activation energy of the observed interlayer and $R_{a}$ represents the universal gas constant ($R_{a}=8.3144621\;J/mol\;K$). However, the application of this new model becomes essential only when the tested temperature is found to be substantially below *T_g_*.

#### 2.1.2. Effective Thickness Methods

The simplified method according to European EN 16612 uses the concept of “effective thickness” [12]. This method separates the equivalent thickness for calculating deflection due to bending from the equivalent thickness for calculating stresses. The equivalent thickness for calculating bending deflection is given by Equation (10) where *ω* is a coefficient with a value between 0 and 1. When the value of *ω* is 0, it is assumed that there is no shear transfer between glass plates. A value of *ω* = 1 implies full shear transfer where the foil is fully effective; *h_i_* is the thickness of glass ply (Figure 1a), and *d_i_* is the distance of the mid-plane of the glass ply *i*, respectively, from the mid-plane of the laminated glass (12).

The value of *ω* is given regarding the stiffness family and load condition of the interlayer. It is presented in Table D.3 from the [12]. Interlayers can be associated with stiffness families 0, 1, and 2 which determine coefficient *ω* as follows: for family 0, *ω* = 0 (for any load condition, no shear transfer between glass plates), for family 1 (*ω* = 0–0.3) and family 2 (*ω* = 0–0.7) exact value of *ω* can be read according to the load condition to which the calculated glass is exposed. In table D.2 of EN 16612 [12], there is an overview of load scenarios associated with the duration of the load and temperature. It must be noted that the coefficient *ω* in the EN 16612 [12] standard is more precisely defined compared to the prEN 13474 [18]. More options are offered in the current version of European standards for evaluating coefficient *ω*. Nevertheless, comparing the accuracy of defining foil properties in EN 16612 to those in Trosifol^®^ technical data [5], it can be noticed only a few load durations with the corresponding temperature that very imprecisely approximate the properties of the interlayer. Trosifol^®^ technical document includes relaxation shear modulus for different temperatures. The UltraClear foil was considered since it was used in the experiment. In the technical data, the value of shear modulus for a load duration of 1 s decreases drastically with temperature raise (for example: for −20 °C, G = 250 MPa, and for 25 °C, G = 2.7 MPa). Applying double interpolation (temperature and time), obtained more precise values of shear relaxation modulus can be obtained. Trosifol^®^ also offers datasheets with values of coefficient *ω* [30], considering the version of European standards [31] used to present composite behavior according to Eurocode standards.
(10)$$t_{ef,w}=\sqrt[{3}]{\overset{n}{\underset{i=1}{\sum }}h_{i}^{3}+12\;\omega \overset{n}{\underset{i=1}{\sum }}\left(h_{i}\cdot d_{i}^{3}\right)\;}$$
where *h_i_* is the thickness of the glass panes and *d_i_* is the distance from the mid-plane of laminated glass (Figure 5).

The version of European standards EN 16612 [12] and EN 16613 [31] changes the approach to the calculation of laminated glass. The equivalent thickness method which is related to stiffness families and load conditions with default shear transfer coefficients is disregarded to direct use of interlayer properties in FEM analysis. Viscoelastic models based on the Prony series are considered for finite element calculations [32].

Methods proposed by Galuppi et al. [13] for effective thickness evaluation will be presented in this study: Wölfel–Bennison and the enhanced effective thickness approach. In Equation (11) can be seen how the Wölfel–Bennison approach calculates the thickness using *h*_1_ and *h*_2_ as individual glass plies thicknesses. Figure 5 illustrates a composite of two glass layers and a polymeric interlayer. The beam length l, width *b*, and thickness for each layer: *h*_1_ and *h*_2_ for glass and t for interlayer are the physical values needed for laminated glass definition. Mechanical characteristics of laminated glass are expressed with *E* for Young’s modulus of glass and *G* for the Shear modulus of the polymer. The non-dimensional coefficient *Γ* = 1/(1 + Κ) ∈ (0,1) takes into consideration the capability of the PVB foil to transfer shear stress between the glass plies. For the determination of *Γ* (Equation (12)) a strong approximation is made by using the universal value of *β =* 9.6 for laminated glass. In the principle of virtual work, another coefficient is found Κ (Equation (13)) where the intermediate layer of thickness *t* is defined through the shear coefficient of the intermediate layer χ  = *t*^2^/*H*^2^ and bending stiffness *Bs = EA^*^H*^2^, where *A** is the applicable cross-section area *A* = A*_1_*A*_2_/(*A*_1_ + *A*_2_) and *H = t* + (*h*_1_ + *h*_2_)/2.
(11)$$h_{eff;w}=\sqrt[{3}]{h_{1}^{3}{+h}_{2}^{3}+12\Gamma \frac{h_{1}h_{2}}{h_{1}{+h}_{2}}H^{2}}.\;$$
(12)$$\Gamma =\frac{1}{1+\beta \frac{tE}{{Gbl}^{2}}\frac{A_{1}A_{2}}{A_{1}{+A}_{2}}}.\;$$
(13)$$Κ=\beta Bs\frac{\chi }{Gbtl^{2}}$$

Equation (14) presents deflection-effective thickness determination by the Enhanced effective thickness (EET) approach for the 1D (beam) case. The non-dimensional weight parameter *η* (Equation (15)) serves a similar role to that of *Γ* in the Wölfel–Bennison approach and takes into consideration the capability of the PVB foil to transfer shear stress between the glass plies. Furthermore, a moment of inertia for the monolithic limit (Equation (15)) *I_tot_ = I*_1_ + *I*_2_ + *A^*^H*^2^, and the value for *I_S_ = A^*^H*^2^/*b* (Equation (14)). Shear coupling parameter *Ψ* takes into consideration the boundary and load conditions for the most common cases of the design practice [13]. In the case of simply supported beams under uniform load, the results given by both of these analytical methods match perfectly. This is because the Wölfel–Bennison approach is based on using the universal value *β =* 9.6 that, according to Wölfel’s theory, applies only to the scenario of simply supported beams under uniformly distributed load. The Wölfel–Bennison method yields insufficiently accurate results when changing boundary and load conditions in comparison with the EET method. Coefficient *Ψ* can be calculated using Equation (16) for any glass element boundary conditions, subjected to any load condition. Coefficient *Ψ* can be determined from Table 1 in [13] for glass beams (one-dimensional case) and Table 2.1 and 2.2 in [13] for glass plates (two-dimensional case); in addition to multi-layered laminated glass [33], curved laminated glass [34], and cantilevered laminated glass [35]. For the case of the simply supported beam under uniform load, the expression for *Ψ* = 168/(17*xl*^2^) was chosen.
(14)$${\hat{h}}_{w}=\frac{1}{{\left(\frac{\eta }{h_{1}^{3}+h_{2}^{3}+12I_{s}}+\frac{1-\eta }{h_{1}^{3}+h_{2}^{3}}\right)}^{1/3}}$$
(15)$$\eta =\frac{1}{1+\frac{Et}{Gb}\frac{I_{1}+I_{2}}{I_{tot}}\frac{A_{1}A_{2}}{A_{1}+A_{2}}\Psi }$$
(16)$$\Psi =\frac{\int_{\Omega }p\left(x\right)g\left(x\right)dx}{\int_{\Omega }g_{\prime }{\left(x\right)}^{2}dx}$$

### 2.2. Experimental Research 

The experiment was performed to obtain deflection data of laminated glass due to long-term loading at different temperatures. Since the load duration and temperature are variable, the application of uniform load to the panel, self-weight was used (250 kg/m^2^). During the installation, several things needed to be secured. It was necessary to avoid (as much as possible) sudden movements of the glass to prevent a change of its initial shape before starting the measurement. The measurement needed to start from the moment the load acts on the glass. It was necessary to ensure that no deflection occurs before the start of the measurement (e.g., with additional pads) (Figure 6a). After the instant deflection was detected, measurements were executed within intervals of 1 min. The glass supports were placed along the shorter sides of the glass panel, freely supported by vertical pads and not clamped anyhow (Figure 6b). Expected deflections and shape changes were dependent on the installed foil. The measured parameters were time, maximum deflection, and glass temperature.

The panel dimensions were 500 mm in width and 3000 mm in length. The laminated glass panel was composed of two 5 mm annealed glasses bonded by a transparent Trosifol^®^ PVB UltraClear interlayer with a thickness of 0.76 mm. The technical data of Trosifol^®^ was considered for the mechanical properties of the interlayer [5]. The effective panel span between vertical supports was measured at 2800 mm with an overhang of 100 mm on each side. The load on the panel was only its self-weight. The total test measurement took 2 days with variable indoor temperature conditions. The real-time temperature conditions were observed, during which the inside temperature for 2 days was around 20 °C with a peak of 29.4 °C after the first day of measurement. It is important to notice that most research on laminated glass is performed at a controlled temperature when the only variable is time, and the relaxation properties of shear depend only on the load duration. During this test, two relatively serious rises and falls in temperature occurred, which are expected to happen during the exploitation time as well.

Glass assembly and the installation of all test devices with additional pads that prevent deflections before the start of the measurement were accurately executed (Figure 6a). The deflection measuring device was WA-T: Inductive Displacement Transducer with highly reliable measurement results [36]. The nominal measuring range of this transducer is up to 100 mm. The principle of device measurement is an inductive quarter bridge circuit based on the differential inductor principle which is completed internally to form a full bridge circuit. 

The measurement started before the extra pads were pulled out. Extra pads were carefully pulled out so that the glass does not vibrate. In the data measurement, interruptions did not occur.

The test results are presented for the 48-h period when the temperature changes presented a great impact on the deformation curvature (Figure 7).

### 2.3. Numerical Modeling

Currently available analysis software greatly simplifies the calculation of laminated glass and the prediction of the glass behavior. Since it is difficult to predict the conditions to which the glass will be exposed during its time of use, based on the performed experiment, the idea was to compare results from software and compare the measured results from the experiment. Numerical analyses were carried out in the following software: SCIA Engineer 21.1. [16], Dlubal RFEM 5.27 [15], Abaqus SIMULIA [21].

SCIA Engineer Addon for glass calculates the effective thickness based on the prEN 13474 [18] regarding the shear stiffness of the interlayer taking into account the duration of the load and the temperature very roughly. The static analytical-numerical calculation gives unnecessarily large glass thickness with a large degree of confidence. Properties of the interlayer are defined through the interlayer stiffness families (0, 1, 2, 3, and 4) in SCIA Engineer. After Addon calculates the effective thickness, it is necessary to proceed with the FEM static analysis on the 2D plate with a relevant thickness within SCIA Engineer. Coefficients *ω* are equal to those given in Table 12 from the [12]. The effective thickness obtained in SCIA Engineer software is additionally reduced according to prEN13474 [18] standards for the interlayer thickness (0.76 mm). Since the thickness of the foil is not included in the effective thickness of the composite, the deflections increase. It should also be observed that in addition to the mentioned effective thickness reduction, SCIA’s calculation is based on old regulations prEN 13474 [18] that take into account the properties of the interlayer even more imprecisely compared to the European standard EN 16612 [12].

Dlubal RFEM [15] offers 2D calculation (with or without shear coupling of layers) and 3D calculation. The difference between these two methods is that 2D calculation is using plate theory within each layer is defined as a surface element, and 3D calculation is carried out by finite element modeling of solid. To simulate the deflection of laminated glass panels due to changes in foil properties, it is recommended to use 3D calculation. For the shear modulus, because load duration is not considered yet in RFEM, it is possible to enter the correct values from the material library. Alternatively, elastic (E) and shear (G) modulus could be set manually so that they match the real modulus depending on the load duration and the glass temperature. In this research Trosifol^®^ technical document was used as a reference and according to the values for Trosifol^®^ Clear foil, the numerical calculation was provided [5]. Temperature curves for the Shear relaxation modulus (Figure 8) and the Young relaxation modulus (Figure 9) dependent on load duration are presented in Figure 8. The study from Hána et al. [37] provides RFEM analysis of laminated glass that is in good agreement with the enhanced effective thickness (EET) approach and Wölfel–Bennison approach in the case of a simply supported beam under the uniform load [13,38]. Another study from Gwóźdź et al. [39] suggests non-linear analysis to be performed in the RFEM in the case of plates with linear supports on all four edges. 

A more detailed static FEM analysis was performed using Abaqus SIMULIA. The numerical model from three solid elements was defined and the material properties of glass and viscoelastic foil were assigned. For viscoelastic behavior, the thermo-rheological simple (TRS) was chosen. Further, the temperature effects that define the shift function (WLF) were included for Ts = 50 °C and recommended values for coefficients C_1_ and C_2_ [14]. Plate elements of 2800 × 500 mm were only vertically supported on both edges, with meshing element dimensions 0.05 × 0.05 m. The loading was defined as a value of uniform pressure load 0.25 kN/m^2^. Finite element modeling analysis in Abaqus SIMULIA was performed only for one point (1500 min and 29.4 °C), while in SCIA Engineer and RFEM software 45 static analyses (points) were observed to draw the deformation curvature over 2800 min.

## 3. Results

In the following section, results will be presented for the deformation measured in the experiment, deformation from the analytic calculations, and deformations obtained from FEM analyses in software. 

The results from the measurement that took 48 days are presented in Figure 10. It is noticeable in the measurement data results (Figure 10) that, even though there is a temperature difference during the time, the glass has relatively consistent deformation. The instantaneous deflection of 27 mm was measured in the experiment that corresponds to the full composite behavior (G → ∞). Very quickly, inside the first minute, creep deformation of 4 mm was measured additionally with total deformation propagation of 35 mm within 2 h. In the time of 2.5 h, deformation stayed around a constant value until the temperature rise occurred from 20 °C to 29.4 °C in the middle of the measuring time. The peak in temperature that occurred after one day of measuring (1500 min) caused deformation to increase to the maximum of 37.9 mm, which stayed constant till the end of the measuring time due to the temperature drop and shear stiffness increase. 

### 3.1. Analytical Calculations

The temperature peak that occurred in the middle of the measuring time was the mismatching point for the comparison of measurement data and design analysis calculations performed in this study. A simple analytical design approach defined in EN 16612 [12], Wölfel–Bennison approach [2], and a more detailed analysis of laminated glass using the TTS principle [13] implemented in the Wölfel–Bennison approach was performed in this study. Depending on the chosen method, according to Equation (17) for the simply supported beam, the deflection is calculated using the moment of inertia that has been calculated for the effective glass thickness. The results were compared. Since the results gained from both the Wölfel–Bennison and EET approaches match perfectly in the case of simply supported beams under uniform load, no additional comparison of the implemented TTS principle in the EET method was carried out.
(17)$$w=\frac{5}{384}\frac{q\cdot l^{4}}{E\cdot I}$$

The behavior of the laminated glass, more precisely, the deflection curve behavior through time was obtained from different laminated glass design approaches. European standards contain a certain amount of safety that is considered through the calculation of structural elements in any material. The deflection curve of the interlayer obtained by calculation according to European standards is shifted by a degree of safety 2.5 to the curve drawn from measured deflections with a total of 98 mm. 

The equivalent thickness approach given by the European standards is proven to be very conservative which leads to unnecessarily large deflection (Equation (10)). Omega coupling values, *ω*, are given in EN16612 [12]. Trosifol^®^ technical data [30] offer tables with *ω* coupling values for diverse types of interlayers. Values for UltraClear foil were used for the calculation and display of the deformation curve according to EN 16612 and EN 16613 (Figure 11a).

An analytical formulation for the calculation of the effective thickness within the Wölfel–Bennison approach (Equation (11)) (using Trosifol^®^ UltraClear data sheet for the Shear relaxation modulus) at the peak temperature of 29.4 °C and time point of 1500 min resulted in a gap of 6.51 mm from the measured data which makes 15% error in the result (Figure 11b). Results are sufficiently accurate with the measured deformations when the temperature is close to constant (0–1000 min in Figure 9), but the problem occurs when there is a temperature difference, as shown in Figure 10. The WLF model was used for the correction of the relaxation modulus provided by Trosifol^®^. Since the temperature rises from 20.8 °C to 29.4 °C in a relatively short time, foil viscoelastic properties have to be included. The analytical approaches from Galuppi et al. [13] rely on the definition of a secant modulus (time and temperature-dependent). Thois is well presented in Figure 10 (1500 min), where the deformation curvature is in slight growth during the measurement, but from the analytical results (Figure 11), the curve is much steeper. The analytical formulation gives unrealistic high deformation values and for that reason, the WLF method was used to solve the problem of interlayer mechanical properties. Reference temperature was chosen to be *Ts* = 50 °C and WLF constants *C_1_* = 8.86 and *C*_2_ = 101.6 as recommended in [14]. Furthermore, *C_1_^i^* and *C_2_^i^* were calculated using Equations (7) and (8), where *T_i_* was listed from measuring data for every minute. Finally, the TTS shift factor was calculated (Equation (6)) based on which the Relaxation modulus from Trosifol^®^ was corrected (Equation (3)). With a new Shear modulus of the interlayer, the Wölfel–Bennison approach proceeded for obtaining deformation. For 2800 points deformation was calculated and the curvature was drawn. In this way, the results have less than a 3% deviation from measured data. The logical input to the code was that the deformation curve can’t reverse under constant stress (which would happen in analytical results if the temperature drops).

Inside this study, laminated glass deformation was analytically solved using the WLF approach using shear relaxation modulus from Trosifol^®^ UltraClear (Figure 8). Relaxation modulus values given in the Trosifol^®^ datasheet were linearly interpolated to determine more precise values for the shear relaxation modulus for the exact time and temperature. Trosifol^®^ datasheet is determined by dynamic mechanical analysis following EN ISO 6721. The sample storage temperature was 23 °C before measurement. The four-point bending test [40] proved to be an adequate nondestructive method for determining relaxation shear modulus. Values from the Trosifol^®^ table are in good agreement when the temperature curves are close to each other, which can be seen in Figure 11b. 

The experimental campaign in this study adopts two variables, time and temperature, under the static loading. Thus, additional correction of relaxation modulus from Trosifol^®^ is required. Additional time-temperature superposition (TTS) is needed when there is a change in temperature during the time, as happened at the end of the first day of measurement in this study. The shift of modulus from different temperature curves was made to fit the reference temperature. Using the TTS principle on wider time domains as presented in this study, shift factor *a_T_* translates horizontally the Young and shear modulus of foil from the measured temperature in exact time to the reference temperature with the new time *t/a_T_*. When presented graphically *a_T_* (Figure 4 and Figure 12) through the time of the experiment, it can be seen the curvature is mirrored to the temperature curvature, thus reducing the impact of temperature changes on the mechanical properties of foil. For verification of the credibility of the shift factor, a simple check can be conducted when *T = T_0_* –> *log a_T_ =* 0 → *a_T_ =* 1 and *t/a_T_ = t*, which proves the time stays unaltered when the temperature curve matches the reference temperature. 

WLF factors for *C_1_* and *C_2_* were calculated for every minute, based on which the shift factor was obtained. For example, in 1500 min and the temperature peak of 29.4 °C, *C*_1_^29.4^
*=* 11,11 and *C*_2_^29.4^
*=* 81.00. After the time correction is made using reference temperature *T_0_ = T_S_ =* 50 °C, with the corresponding WLF factors *C*_1_*^s^ =* 8.86 and *C_2_^s^ =* 101.6 as recommended in [14], new shear modulus *G (t/a_T_, T_0_)* were calculated. When the mastercurve is obtained and the constants *C*_1_ and *C*_2_ for each time and temperature point are known, all mastercurves in the range of validity of the WLF model could be obtained. Thus, the time and temperature viscoelastic modulus of the material is completely defined [14]. Additionally, deformations of the laminated glass were calculated using the Wölfel–Bennison with a new shear relaxation modulus. In that way, obtained deformation fits deformation measurement results within less than *3%* (Figure 11c).

A comparison of applied analytical approaches was presented in Figure 13. Deformations using the TTS principle match the measured data the best since both—time and temperature were taken into account. Using the TTS principle, additional attention was paid to ensure that the deformation could not be smaller even if the stiffness of the glass increases under lower temperatures (as happened during the second day of measurements), remaining constant or slowly propagating over time, depending on the conditions, until the foil loses its shear stiffness and the plies start to act separately. The TTS principle is in the best agreement with the measurement data within 3% accuracy and the worst is EN 16612 with 50 mm larger deflections from measured ones. 

### 3.2. Numerical Analysis

Two software programs for structural analysis were used to obtain deformations: SCIA Glass Addon (Figure 14) and Dlubal RF GLASS (Figure 15). The numerical model is a double-sided hinged plate (2.8 × 0.5 m) with meshing elements 0.05 × 0.05 m. The loading was defined as a value of uniform surface load 0.25 kN/m^2^. The performed analysis was static.

SCIA Engineer has a glass calculation add-on (Figure 14) based on calculations from pre-norms prEN 13474 [18]. Mechanical properties of glass and interlayer material were modeled within glass addon and linear static analysis was run. Properties of the interlayer were defined tabularly by choosing the interlayer stiffness families (0, 1, 2, 3, and 4). The calculation of the effective thickness of the assigned numerical model is given according to pre-norms and that calculated thickness is the entry in the numerical calculation. Regarding the shear stiffness of the interlayer, SCIA Engineer takes into account the duration of the load and the temperature very roughly. This analytical-numerical calculation gives unnecessarily large glass thickness with a large degree of confidence (Figure 16a). Choosing Family 1, the value of coefficient *ω* was set to 0.1. SCIA Engineer obliterates the foil thickness through Equation (10) using the distances of the mid-plane of the glass plies *i* respectively and not including foil thickness. The value of the effective thickness in the glass addon is 2–7% reduced to the current European standards, resulting in a 7% deviation in results and a total of 105 mm in deflection. SCIA Engineer gives an error of 64% from the measured deformation in the experiment. (Figure 17).

The value for the shear modulus in the formulas was obtained from linear interpolation of the given shear relaxation modulus inside of the Trosifol^®^ datasheet and manually entered into RF GLASS (Figure 15). Design approaches that are incorporated into the software Dlubal are in good agreement with the analytical Wölfel–Bennison approach [2] proving the accuracy of the Wölfel–Bennison approach with the FEM analysis of solid elements (Figure 16b). 

Thermo-rheological temperature dependence for time domain viscoelasticity was defined from the suboptions menu (TRS) inside of the material definition for the PVB interlayer. Shift function approximation was defined as Williams–Landel–Ferry (WLF). Reference temperature theta Θ was chosen at 50 °C as suggested by [14] and the WLF factors *C*_1_*^s^* = 8.86 and *C*_2_*^s^* = 101.6. Young modulus and Poisson’s ratio for the elastic properties of foil were selected for the exact time and temperature (1500 min and 39.4 °C). Interpolated elastic relaxation modulus was obtained from the Trosifol^®^ table E = 0.591 MPa and Poisson’s ratio 0.495. The results are shown in Figure 18. Results fit the measurement data with almost 100% accuracy. 

## 4. Conclusions 

The demand for products made of laminated glass is growing every day, so it is necessary to adjust the design methods to optimize the utilization of the material. In this study, an attempt was made to simulate the actual behavior of the laminated glass at a given moment to expand it to its lifetime use. In this paper, the viscoelastic behavior of the PVB interlayer was observed in dependence on time and temperature variables. The application of the TTS and WLF model was analyzed and compared to the currently available standards for laminated glass design. 

The relaxation modulus at different temperatures was adopted from the Trosifol^®^ catalog E(t) and G(t) with the additional linear interpolation. The WLF model was used for the correction of the relaxation modulus provided by Trosifol. The shear modulus in the producer’s technical data is accurate for the constant temperatures. The problem occurs when there is a change in the temperature. The reason for this lies in the inability of the interlayer to follow the glass due to the memory effect. The analytical approaches based on effective thickness only consider the time and temperature in the current moment. The analytical formulation gives unrealistic high deformation values and for that reason, the WLF method was used to solve the problem of interlayer mechanical properties. For the WLF model constants, *C_1_^WLF^* = 8.86, *C_2_^WLF^* = 101.6 and the reference temperature *Ts =* 50 °C most accurate results were obtained in comparison to the measured data. The deformation curvature change that occurs within analytical approaches was well maintained using the shift factor *a_T_*. 

Compared to the measured data, the error generated in design approaches followed by European standard was approximately 61% for analytical calculation and 64% in FEM analysis proceeded in glass addon within SCIA Engineer software. Another analytical approach, i.e., Wölfel–Bennison approach, was in good agreement with the measurement when the temperature was constant, but the temperature rise was the breakdown point with an error of 15%. The same results are gained using an enhanced effective method for analytical calculation. The RF GLASS 3D analysis within RFEM Dlubal software was found to be 100% accurate with Wölfel–Bennison approach and the enhanced effective method when the mechanical properties of the foil were manually defined. In SIMULIA, the viscoelasticity of the PVB interlayer was defined through thermo-rheological temperature dependence (TRS) and found to be in 100% agreement with the experiment. 

## 5. Discussion 

In future research on laminated glass plates, dynamic loading will be observed due to the additional parameters that should include instant deformation, creep deformation, and instant recovery deformation that is memorized into the foil when changing the direction of the loading during the time. Load duration and temperature changes under the changing pressure are difficult to implement into the laminated glass behavior without simulating a longer period of laminated glass usage. Finally, the potential to prolong the laminated glass’s lifetime can be proposed. 

## Figures and Tables

**Figure 1 polymers-14-04402-f001:**
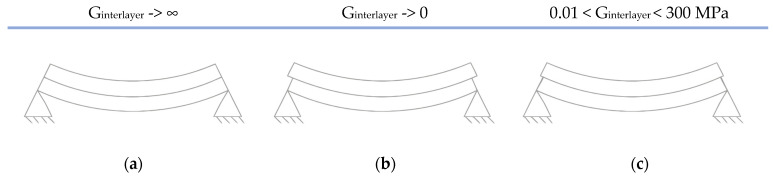
Laminated glass composed of two plies under bending: (**a**) monolithic limit; (**b**) layered limit; (**c**) intermediate configuration.

**Figure 2 polymers-14-04402-f002:**
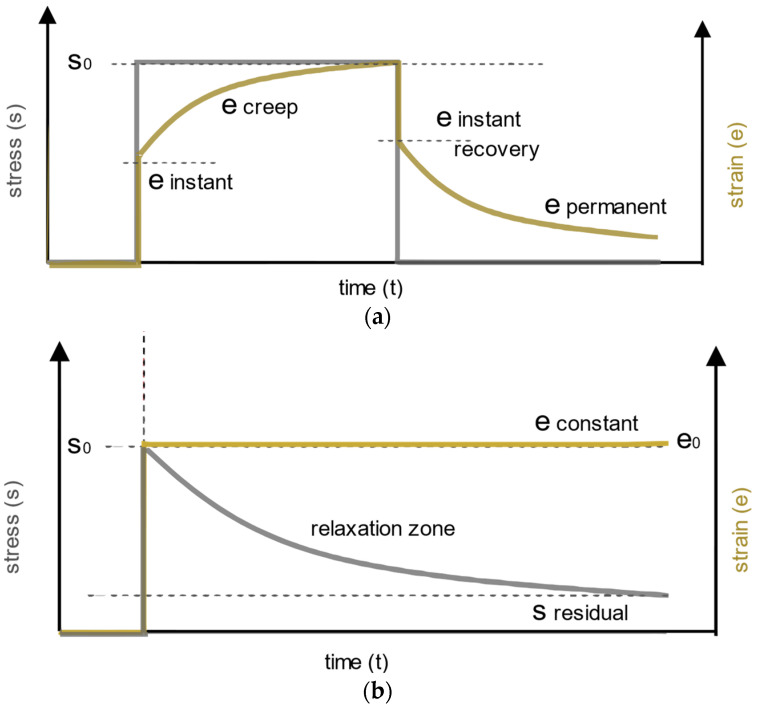
Stress and strain diagrams of viscoelastic material: (**a**) when constant stress is applied; (**b**) constant strain is applied. Adapted from [23], Elsevier, 2021.

**Figure 3 polymers-14-04402-f003:**
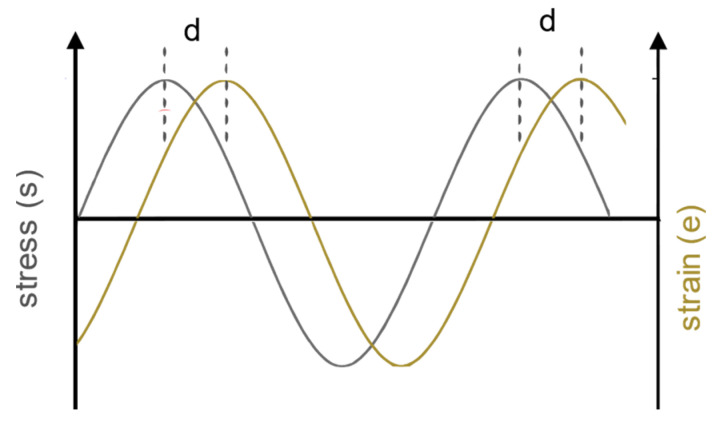
Temporal offset of stress and strain diagram of a viscoelastic material under dynamic load. Adapted from [23], Elsevier, 2021.

**Figure 4 polymers-14-04402-f004:**
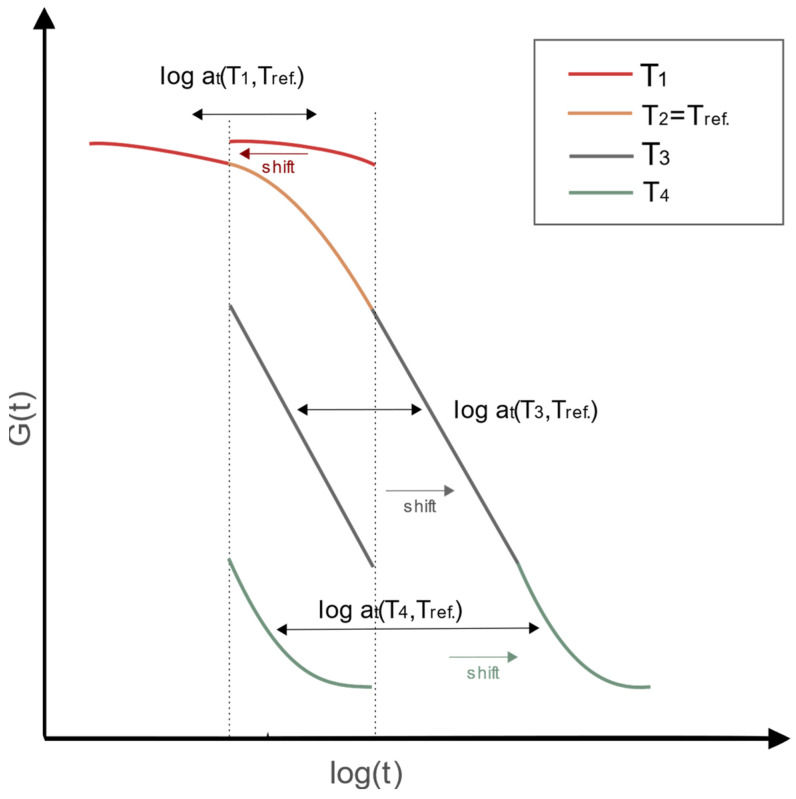
Horizontal shift of temperature curves to match reference temperature curve, *T_1_ < T_2_ < T_3_ < T_4_*.

**Figure 5 polymers-14-04402-f005:**
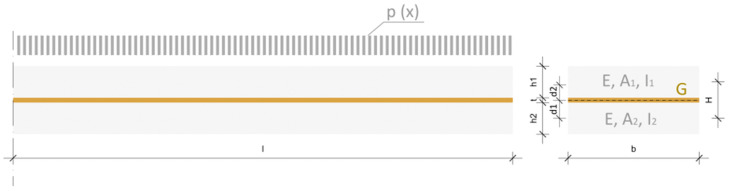
Laminated glass composed of two glass plies and polymeric foil.

**Figure 6 polymers-14-04402-f006:**
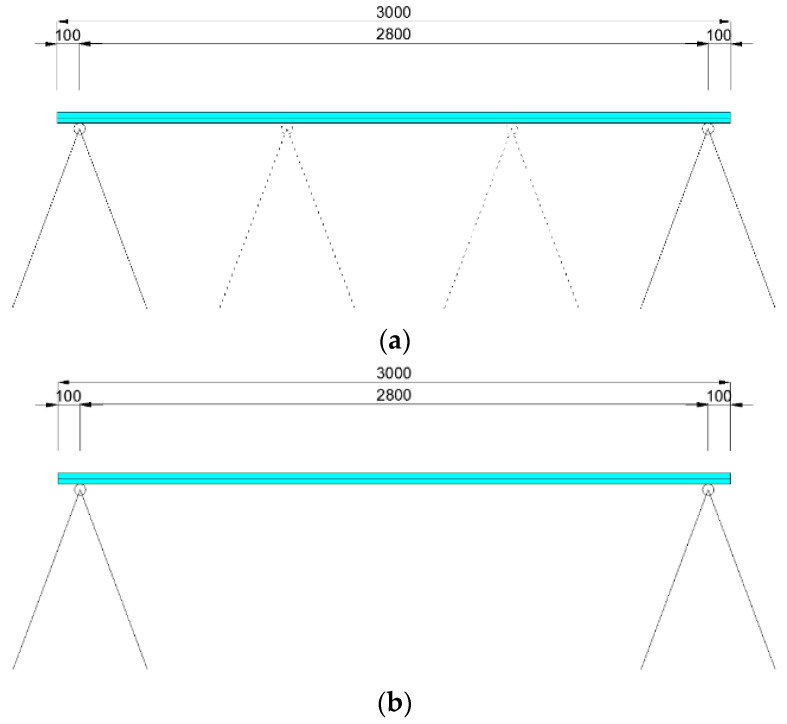
Static scheme of the laminated glass: (**a**) during the mount, (**b**) during the measurement.

**Figure 7 polymers-14-04402-f007:**
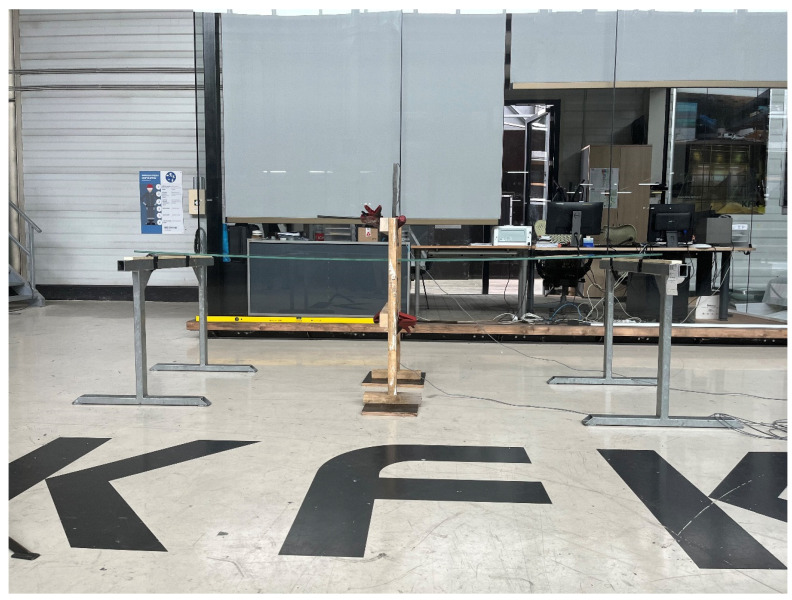
Displacement measurement of laminated glass in KFK d.o.o. (Zagreb, Croatia).

**Figure 8 polymers-14-04402-f008:**
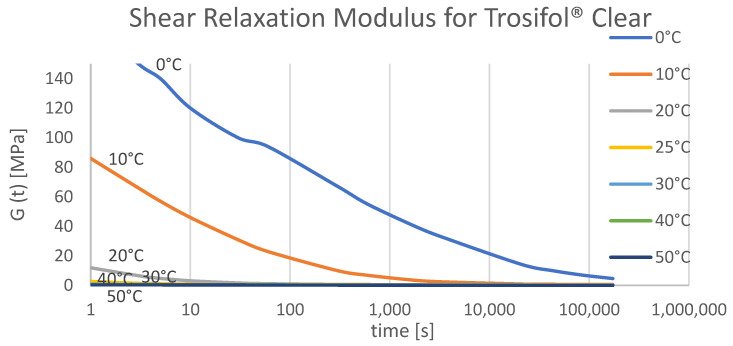
Shear Relaxation curves for PVB Clear from Trosifol^®^ data sheet at different temperatures.

**Figure 9 polymers-14-04402-f009:**
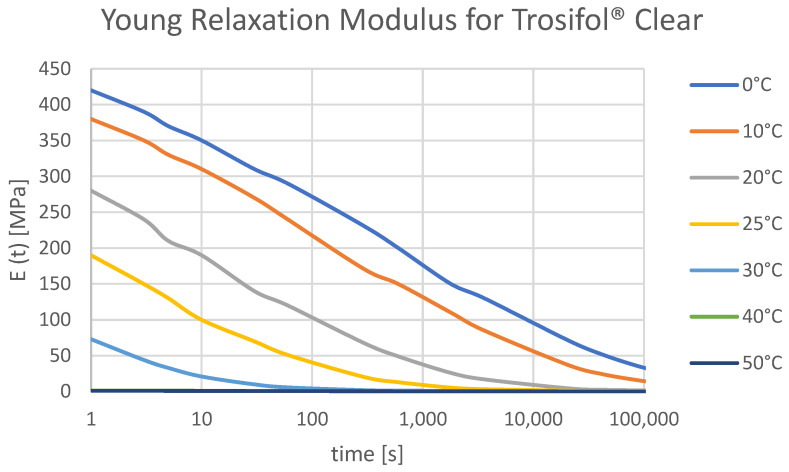
Young Relaxation curves for PVB Clear from Trosifol^®^ data sheet at different temperatures.

**Figure 10 polymers-14-04402-f010:**
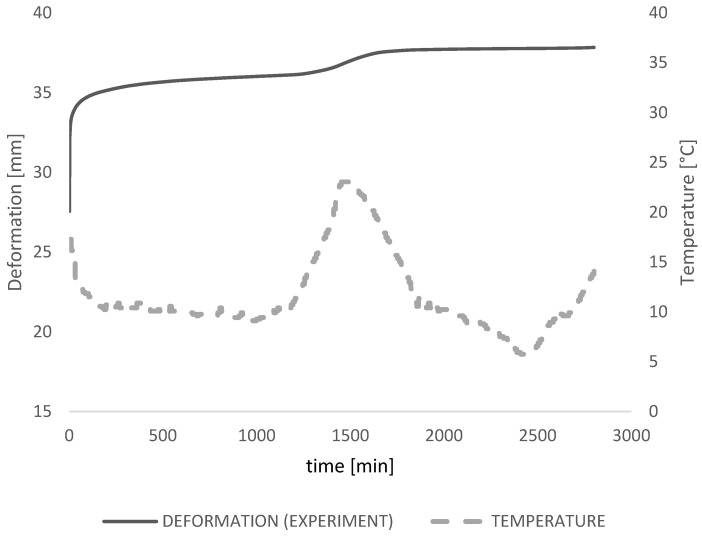
Measured values from the experiment for the deformation and the temperature for 48 h.

**Figure 11 polymers-14-04402-f011:**
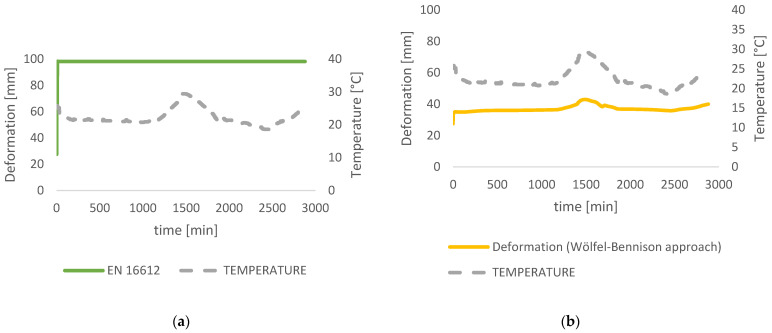
Deformation from different glass design analytical approaches: (**a**) norm EN 16612; (**b**) Wölfel-Bennison approach; (**c**) Enhanced effective thickness approach; (**d**) Effective thickness approach combined with TTS correction of Shear Modulus *G*.

**Figure 12 polymers-14-04402-f012:**
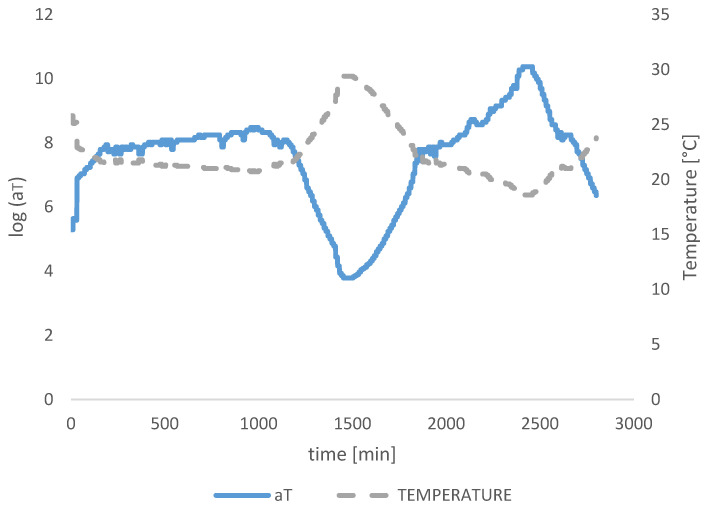
Shift factor and temperature diagram in time.

**Figure 13 polymers-14-04402-f013:**
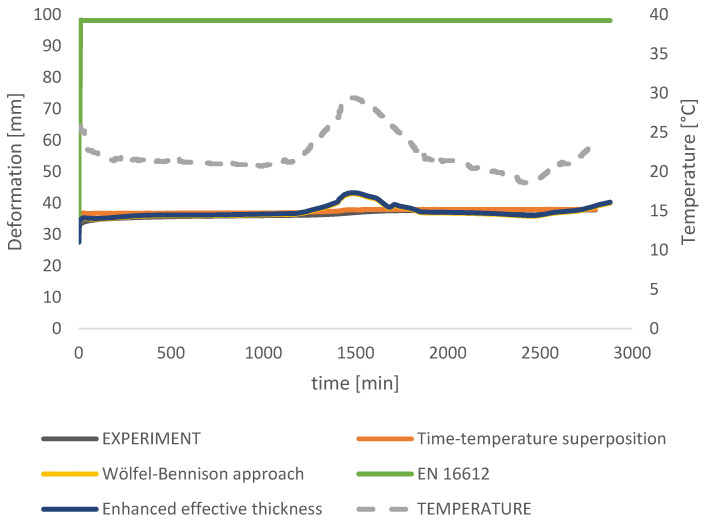
Deformation comparison of different glass design analytical approaches with measured data: EN 16612; Wölfel-Bennison approach; Enhanced effective thickness approach; Effective thickness approach combined with TTS correction of Shear Modulus *G*.

**Figure 14 polymers-14-04402-f014:**
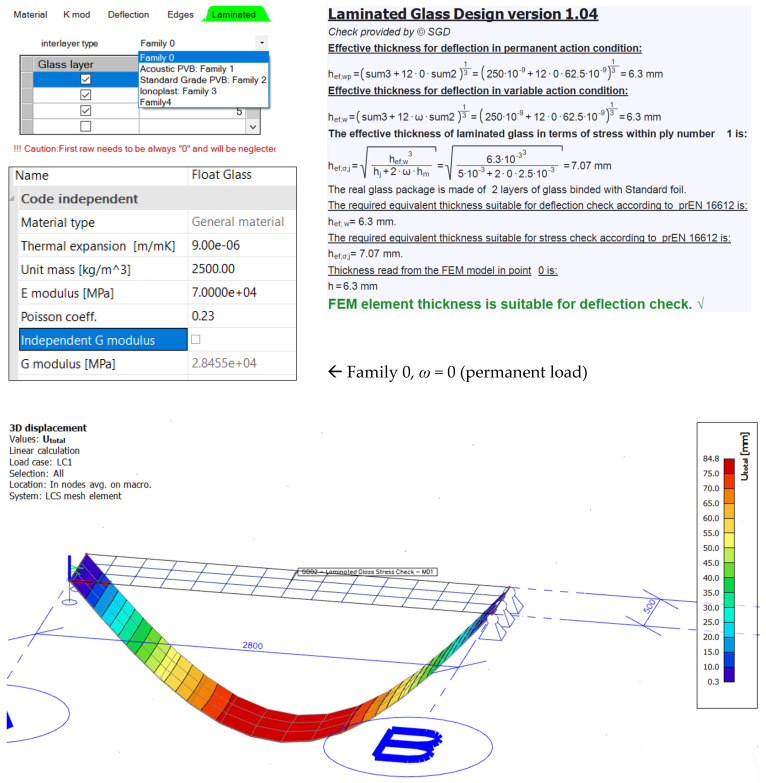
SCIA Engineer—addon for glass design.

**Figure 15 polymers-14-04402-f015:**
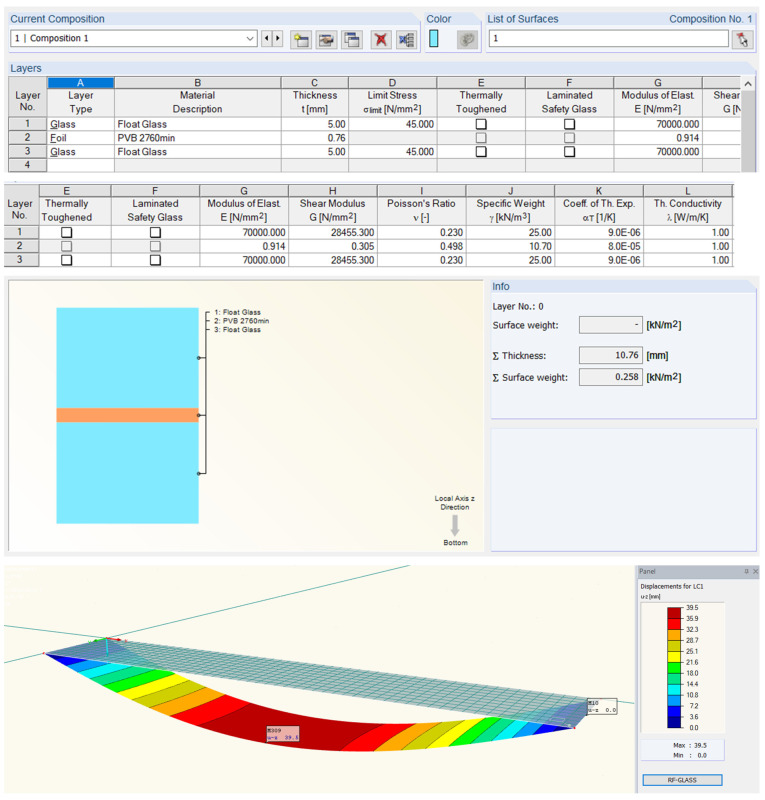
Dlubal RFEM—RF GLASS for glass design.

**Figure 16 polymers-14-04402-f016:**
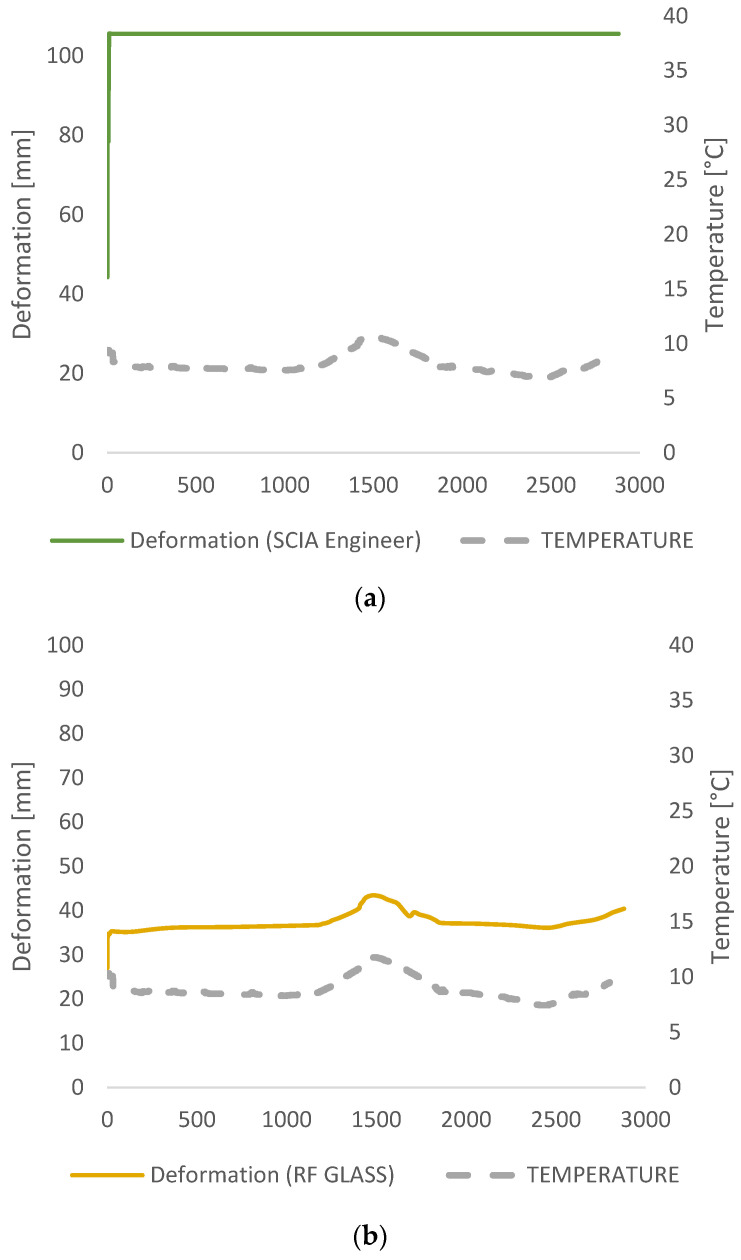
Deformation from different software FEM analysis: (**a**) SCIA Engineer; (**b**) RFEM—RF GLASS.

**Figure 17 polymers-14-04402-f017:**
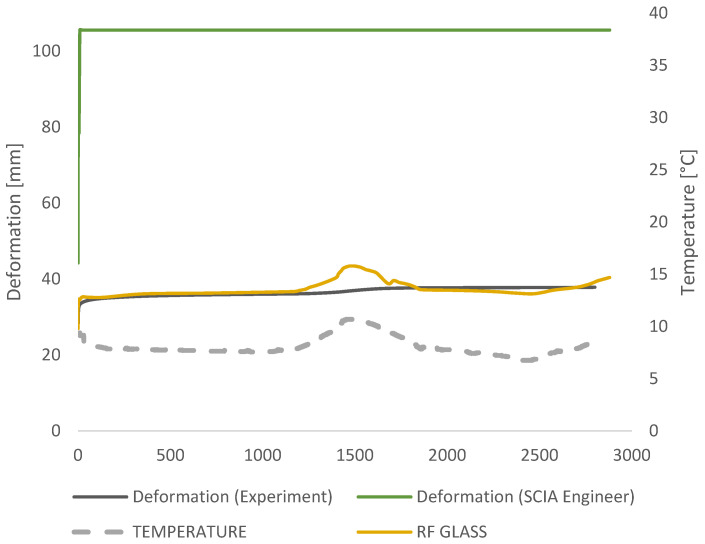
Deformation comparison from different software FEM analysis and measured data from the experiment.

**Figure 18 polymers-14-04402-f018:**
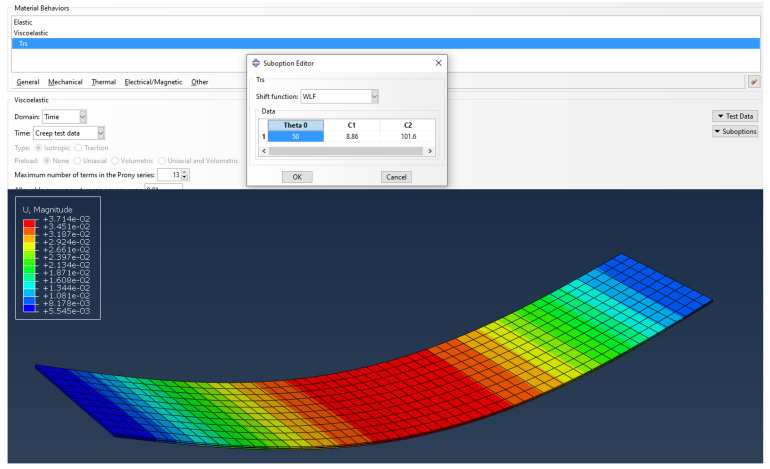
Abaqus SIMULIA FEM analysis of laminated glass.

## Data Availability

The raw data presented in this study are available on request from the corresponding author.

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
