# Peer review of "Influence of PVB Interlayer Mechanical Properties on Laminated Glass Elements Design in Dependence of Real Time-Temperature Changes"

_polymers, 2022, doi:10.3390/polym14204402_

Round 1

Reviewer 1 Report

The paper presents the results of an experimental campaign, and the comparison with analytical and numerical results, on 2-layered simply supported laminated glass beams under self-weight. In the filed of laminated glass, this is not an innovative contribution, since several similar studies has been presented in the last decade. Furthermore, the considered analytical methods are not those currently used for laminated glass: reference is made to an expired standard, while the most modern methods suggested by current norms are not considered). Numerical analyses have been performed by using the same analytical model, so providing trivial results.

Major comments:

·        - Both in the abstract and in the introduction, it is stated that “Most structural engineers are designing laminated glass regardless of the shear coupling of the plies”. This statement is incorrect, since, at least in the last decade, most of the Standards suggests analytical method for the evaluation of the shear coupling provided by the interlayer(s).

·        -Comparisons are made with respect to prEN13474, that is an old norm, that has been replaced by EN16612 three years ago. The comparisons should be performed with reference to the current standard. Furthermore, comparisons should be made also with the Enhanced Effective Thickness approach [Galuppi, et al.. "Practical expressions for the design of laminated glass." Composites Part B: Engineering 45.1 (2013): 1677-1688.], widely used in Europe and suggested by  prEN 19100-2 Eurocode 11 Eurocode 11 — Design of glass structures - Part 2: Design of out-of-plane loaded glass components.

·       - The authors seem to be quite confused about the current standards: EN16612 is not a prEN (since 2019), and it has replaced the old prEN13474.

·       - It is not  clear why the Time-Temperature superposition principle is described as an approach to design laminated glass (introduction, line 100)

·     -   Sect. 2.1 should be divided into 2 subsection: viscoelastic response of the interlayer and Effective Thickness methods. Since this is a Polymer journal  the former part should be probably reduced, since the presented formulation is very standard. On the contrary, the latter part should be enhanced and enriched: a figure showing the relevant geometric quantities (thicknesses, etc) should be added; the mathematical treatment should be better presented (for example, all the quantities appearing in the equations should be defined, see also minor comments), and other methods (as EET) should be included. NOTE: For the case at hand (simply supported beam under a uniformly distributed load), the W-Bennison approach is quite accurate, and results obtained with EET are expected to be almost coincident with those of WB.

·        -The numerical modelling should be better described: which kind of elements and mesh have been used? The adopted mesh is the same for all the considered softwares? What kind of analyses has been run (static/quasi-static…?)How the material properties have been modelled (table of values/ Prony series/…?). Is the interlayer modelled as a full viscoelastic material, or with a secant modulus? All these detailed description should be recorded in Sect. 2.2. Furthermore, the analyses performed with abaqus are not described at the end of sect. 2.

·        -It is completely not clear how the WLF approach (only providing the material properties of the interlayer) has been used to evaluate the composite deflection. This is an important point, that must be clarified.

·      -  It is well known that effective thickness methods does not provide the correct time-dependence of the laminated glass response, because they rely upon the “secant stiffness” approximation, completely neglecting the viscoelastic memory of the polymer. As analytically demonstrated in [Galuppi, and Royer-Carfagni. "Laminated beams with viscoelastic interlayer." IJSS 49.18 (2012): 2637-2645.] the secant stiffness approximation underestimates deflection (for monotonic load). Hence, is not surprising that all the considered ET models underestimate the beam deflection.

·        -It is well known that the model proposed by prEN 13474 is not accurate for the design of laminated glass. This is why this standard has been replaced with the more accurate EN 16612. The comparisons must be made wrt the current norm.

·       - Since SCIA eng is based on the approach by prEN13474, it’s not surprising that the results coincide with those obtained by analytically apply the omega method. Indeed, once the effective thickness is evaluated, the max deflection can be evaluated simply by means of classical beams formula, with no need to a FEM program. This comparison is hence superfluous, and should be removed by the paper.

·       - At the same manner, since RFglass is based on the definition of the shear modulus, it is obvious that the results are in good agreement with the W-Bennison approach.

·     -   In light of the previous comments, the conclusion section should be completely re-written.

·     -   A suggestion: before dealing with dynamics, it would be appropriate to carefully read the (lot of) work done by the group of professors Aenlle and Pelayo on “dynamic effective thickness”.

·       - A lot of relevant literature contribution are missed in the reference list, as for example:

o   Shitanoki, Bennison, and Koike. "A practical, nondestructive method to determine the shear relaxation modulus behavior of polymeric interlayers for laminated glass." Polymer testing 37 (2014): 59-67.

o   Hána, et al. "Current analytical computational methods of laminated glass panels in comparison to FEM simulation." Structures and Architecture: Bridging the Gap and Crossing Borders. CRC Press, 2019. 619-627.

o   Hána, et al. "Advanced computational methods of perpendicularly loaded laminated glass panes." (2019).

o   Gwóźdź, Marian, and Piotr Woźniczka. "New static analysis methods for plates made of monolithic and laminated glass." Archives of Civil Engineering 66.4 (2020).

Minor comments:

·        Introduction, line 36: “Laminated glass is a composite of 2 glass  layers with…”. This sentence should be modified, because in many structural application more than 2 glass plies are needed, also according to the Standards

·        Introduction, line 45: the sentence about “the strength of laminated glass” should be modified. The strength is a material property (of glass, in the present case). It would be more appropriate to talk about the “load bearing capacity” of laminated glass elements.

·        Introduction, line 50: ref. [3] does not consider transverse actions

·        SentryGlas (with only one “s”) is a registered trademark and should be denoted with®

·        EN16612 is not a prEN (since 2019), and it has replaced the old prEN13474.

·        line 110: why the acronym for Temperature-Shift Time is TRS?

·        line 123: again, “strength” is not the correct word here

·        Eq. (12-13): all the quantities appearing in the equation should be defined!

·        Line 267: the sentence “The largest deflection was 267 expected in the first minutes” seems to be not correct. Indeed, due to the interlayer relaxation, the deflection is expected to increase in time.

·        Line 323: it is not clear why “To simulate deflection of laminated glass panel due to changes in foil properties it is recommended to  use 3D calculation

·        Figure 9: to emphasize what happens in the first minutes, the graph should be plotted in log scale. The same holds for Figs. 11, 12, 15, 16.

·        Sect. 3.1: it should be clearly stated how the max deflection has been evaluated, starting from the effective thickness.

·        Figure 14: which load duration has been used for the material definition in Dlubal RFEM (i.e., to choose the value for E, G)?

·    

Author Response

Please find the Response in the word document!

Reviewer 2 Report

The paper is interesting and well-written. I have nothing to add/comment

I could suggest only the following minor change, just for clarity:

paragraph 2 (materials and methods). The description order of "slide free" (shear modulus tends to zero) and "monolithic glass" (shear modulus tends to infinite) is reversed respect to that presented in figure 1 (Fig.1a showing the monolithic glass, Fig. 1b the "slide free"). I just suggest to switch the dscription order in the text, or the Fig.1a-Fig1b

Author Response

(The authors gave the same response as above.)

Reviewer 3 Report

1- Need to add some relevant references to the manuscript published in the Polymers journal. I am not interested to name them.

2- The respected authors need to separate the discussion from the conclusion and present it in a separate section and more explanations, or with the results.

3- Equations 11, 12, and 13 need to be reconsidered. I think from a unit point of view they don't match.

4- Is the horizontal shift for T1 to the reference temperature T2 correct in figure 4? I can not see that clearly.

5- Are Figures 1 and 2 the work done in this paper or did the respected author take them from other references? I think it is needed to mention Ref23 and 24 under the figures.

Author Response

(The authors gave the same response as above.)

Round 2

Reviewer 1 Report

The authors have partially updated the paper by following the reviewer’s comments, so improving the paper quality. However, there are still areas that should be improved before publication:

·        English revision by a native speaker is recommended

·        The paper should be accurately checked, so to correct misprints

·        Math terms should be italicized in the text

·        Lines 237-245: it is not necessary to record the exact values for \omega prescribed by the EN 16612. It should be better to record only the stiffness families, and the range of values for \omega, without citing the load cases (that are not defined in the paper)

·        Lines 263-265: the sentence is unclear and should be rephrased.

·        Figure 5 should be moved at the beginning of sect. 2.1.2. Furthermore, quantities appearing in eq. (10) should be quoted in the picture. Finally, it is not clear what “D” is (is it the plate flexural stiffness? If yes, D_1 and D_2 should be used)

·        Please check eq.(13)

·        Line 281: coefficient \eta for the EET model is usually referred to as “shear coupling parameter”

·        Line 284: H_2 should be changed into H^2

·        H is not defined. It should be also indicated in Fig. 5.

·        Lines 293-295: the formulas recorded below (14-16) only refer to the 1D (beam) case. Please rephrase the sentence.

·        Line 296: the original reference should be cited here.

·        Lines 352: according to lines 237-241, only families 0, 1 and 2 are defined. Please check.

·        Line 357: what the authors means by “pre-European standards”?Line 446: ALL the effective thickness models, not only the one by Galupi et al. rely on the definition of a secant modulus (time- and temperature-dependent). Please rephrase

·        Line 452: please add a reference for the values of the WLF constants.From the answers to the reviewer’s previous comments: As the formula for the maximum deflection of a simply supported beam is “trivial to be shown”, also the FEM analyses of a simply supported beam (made with SCIA) are trivial to be shown.

·        Legend of fig.s 12-13 should be modfied to specify that the TTS is used coupled with the ET approach

·        Lines 362, 429, 431, 586,etc: since the new  Technical Specification CEN/TS 19100- “Design of glass structures” has been recently published, to refer to the EN16612 as “latest European standard” could be misunderstanding.

·        From the answers to the reviewer’s previous comments: it is still not clear which load duration has been used for the material definition in Dlubal RFEM

·        The reference list contains many misprints, as well as partial information, and should be revised.

Author Response

Dear reviewer, please find the responses in the attachment. 

Round 3

Reviewer 1 Report

The paper can be accepted in its current form